# A study of evaluating specific tissue oxygen saturation values of gastrointestinal tumors by removing adherent substances in oxygen saturation imaging

Keiichiro Nishihara[1], Keisuke Hori[1], Takaaki Saito[2], Toshihiko Omori[3], Hironori Sunakawa[1], Tatsunori Minamide[1], Masayuki Suyama[1], Yoichi Yamamoto[1], Yusuke Yoda[1], Kensuke Shinmura[1], Hiroaki Ikematsu[1], Tomonori Yano[1]*

1 Department of Gastroenterology and Endoscopy, National Cancer Center Hospital, Kashiwanoha, Kashiwa, Japan, 2 Imaging Technology Center, FUJIFILM Corporation, Tokyo, Japan, 3 Medical Systems Research & Development Center, Research & Development, Management Headquarters, FUJIFILM Corporation, Tokyo Japan

* toyano@east.ncc.go.jp

**Data Availability Statement:** All relevant data are within the manuscript and Supporting Information files.

## Abstract

### Objectives

Oxygen saturation (OS) imaging is a new method of endoscopic imaging that has clinical applications in oncology which can directly measure tissue oxygen saturation (Sto2) of the surface of gastrointestinal tract without any additional drugs or devices. This imaging technology is expected to contribute to research into cancer biology which leads to clinical benefit such as prediction to efficacy of chemotherapy or radiotherapy. However, adherent substances on tumors such as blood and white coating, pose a challenge for accurate measurements of the StO2 values in tumors. The aim of this study was to develop algorithms for discriminating between the tumors and their adherent substances, and to investigate whether it is possible to evaluate the tumor specific StO2 values excluding adherent substances during OS imaging.

### Methods

We plotted areas of tumors and their adherent substances using white-light images of 50 upper digestive tumors: blood (68 plots); reddish tumor (83 plots); white coating (89 plots); and whitish tumor (79 plots). Scatter diagrams and discriminating algorithms using spectrum signal intensity values were constructed and verified using validation datasets. StO2 values were compared between the tumors and tumor adherent substances using OS images of gastrointestinal tumors.

### Results

The discriminating algorithms and their accuracy rates (AR) were as follows: blood vs. reddish tumor: Y> - 4.90X+7.13 (AR: 95.9%) and white coating vs. whitish tumor: Y< -0.52X +0.17 (AR: 96.0%). The StO2 values (median, [range]) were as follows: blood, 79.3%

**Funding:** The National Cancer Center Research and Development Fund provided support for this study in the form of a grant awarded to TY (29-A-10). FUJIFILM Corporation also provided support for this study in the form of a research grant awarded to TY, oxygen saturation imaging for the clinical study, and salaries for TS and TO. The specific roles of these authors are articulated in the 'author contributions' section. FUJIFILM played a role in the image analysis and oxygen saturation endoscopy system, but no other role in study design, data collection and analysis, decision to publish, or preparation of the manuscript.

**Competing interests:** The authors have read the journal's policy and have the following competing interests: TS and TO are employees of FUJIFILM Corporation. TY also received a research grant from FUJIFILM. Additionally, FUJIFILM provided oxygen saturation imaging and assisted with image analysis. This does not alter our adherence to PLOS ONE policies on sharing data and materials. Under the joint research agreement, KN, HK, TY, TS, TO jointly applied for the findings obtained from this study as a Japanese patent (2019-059812). There are no products in development or marketed products associated with this research to declare.

[37.8%–100.0%]; reddish tumor, 74.5% [62.0%–86.9%]; white coating, 73.8% [42.1%–100.0%]; and whitish tumor, 65.7% [53.0%–76.3%].

## Conclusions

OS imaging is strongly influenced by adherent substances for evaluating the specific StO2 value of tumors; therefore, it is important to eliminate the information of adherent substances for clinical application of OS imaging.

## Introduction

Endoscopy was developed to detect gastrointestinal (GI) cancer at the early stages. This procedure has been improved to visualize the architecture pattern of GI mucosa using high definition, magnification, and image enhancement to achieve precise diagnosis [1–3]. While these advancements improved the detection or the accuracy of depth prediction of early neoplastic lesions, there was no significant progress in the diagnosis of advanced GI cancer with endoscopy. Numerous studies have attempted to elucidate the biological characteristics of cancer or to predict the efficacy of chemotherapy for advanced GI cancer using endoscopic imaging [4–7]. If endoscopy could reflect the biological features of cancers in the human body, it would gain additional value as a functional imaging approach, in addition to its current use in architectural diagnosis. Recently, we jointly developed Oxygen Saturation (OS) imaging (FUJIFILM, Tokyo, Japan), a new methodology of image-enhanced digestive tract endoscopy for the qualitative diagnosis of digestive tumors [8]. This approach is based on the detection of hypoxic areas and thereby characterizing the cancer microenvironment. Pathophysiological findings showed that the hypoxic environment in many solid tumors depends on both cell proliferation and angiogenesis. Tumor hypoxic environment induces hypoxia-inducible factor-1 (HIF-1) overexpression, which in turn accelerates growth, differentiation, metastasis, and infiltration of cancer cells. HIF-1 overexpression is also associated with tumor resistance to radiation and chemotherapy [9–14]. Thus, quantification and visualization of hypoxic regions in solid tumors may be important for predicting the efficacy of anti-cancer treatment.

During OS imaging, two different lasers produce light in the blue spectrum that is used to detect the difference in absorption between oxy- and deoxy-hemoglobin. OS imaging allows relatively easy measurement of tissue oxygen saturation (StO2) levels on tumor surfaces in early stage cancers (Fig 1). However, several challenges are encountered in an accurately diagnosing of advanced cancers because of the background noise that originates from blood and white coating present in advanced tumors. This noise can interfere with measurements of tumor specific StO2 values at the surface of tumors. The aims of this study were to develop algorithms for discriminating between tumors and their adherent substances using white light imaging and to investigate whether it is possible to evaluate the StO2 values in the exposed tumor specifically in OS imaging.

## Methods

### Study samples

We analyzed treatment naive cases of advanced gastric cancers and advanced esophageal cancers with blood and white coating as adherent substances, which were assessed by OS imaging (LASEREO system, EG-L590ZW, Fujifilm, Tokyo, Japan) between April and November 2018.

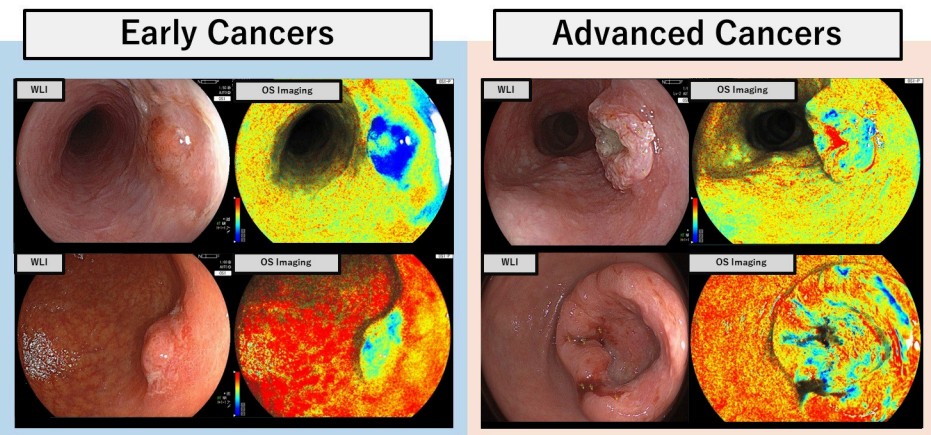

**Fig 1. Oxygen saturation (OS) imaging of early and advanced upper gastrointestinal cancers.** Gastric and esophageal metaplasia images of the early and advanced cancers are shown. OS imaging allowed relatively easy measurement of StO2 levels on tumor surfaces in early cancers. However, advanced cancers tend to be covered on adhered substances, such as blood and white coating, which can interfere with accurate measurements of tumor StO2 values.

OS imaging were extracted from NEXUS system (FUJIFILM Holdings Co., Tokyo, Japan) and fully anonymized and de-identified. The protocol was approved by the Institutional Review Board of National Cancer Center Japan (approval number: 2017-434).

## Image processing of signal intensity using white light imaging

Focusing on blood and white coating at the tumor surface as the main source of noise in OS imaging, we aimed to construct algorithms that could discriminate between adherent substances on the surface of the tumor (blood and white coating) and exposed tumor areas with similar color.

To ensure accuracy, two endoscopists plotted areas considered to represent exposed tumor and areas considered to represent adherent substances at multiple points in each of the images of the 50 advanced cancers using ImageJ (open-source, public-domain image processing software created by Bharti Airtel, Ltd.) [15]. The plotted areas were identified by two-dimensional coordinates. The spectrum signal intensity values (the raw pixel values proportional to the reflection rates of the substances concerned) at those areas were analyzed by custom-made software, developed by FUJIFILM Corporation, and two-dimensional scatter plots were produced (Fig 2). The red, green, and blue (R, G, and B) pixel values of the emitted white light from phosphor illuminated by 445 nm blue laser light were used as spectrum signal intensity values. These spectrum signal intensity values constituted continuous variables and were analyzed using EZR software (http://www.jichi.ac.jp/saitama-sct/SaitamaHP.files/statmedEN.html).

## Constructing discriminating algorithms for tumors vs. their adherent substances

The two G-signal-normalized log signal ratios were the Y axis, which was ln (B445/G) and X axis, which was ln (R/G) the XY two-dimensional scatter plots were produced using data from the X and Y axis. We used the following rationale for the chosen analysis. While hemoglobin absorbance has comparatively little effect on the red spectrum signal (R), its absorbance strongly affects the green spectrum signal (G). As a result, the ratio of ln R/G will be higher at higher hemoglobin concentrations, exhibiting a positive correlation with the hemoglobin

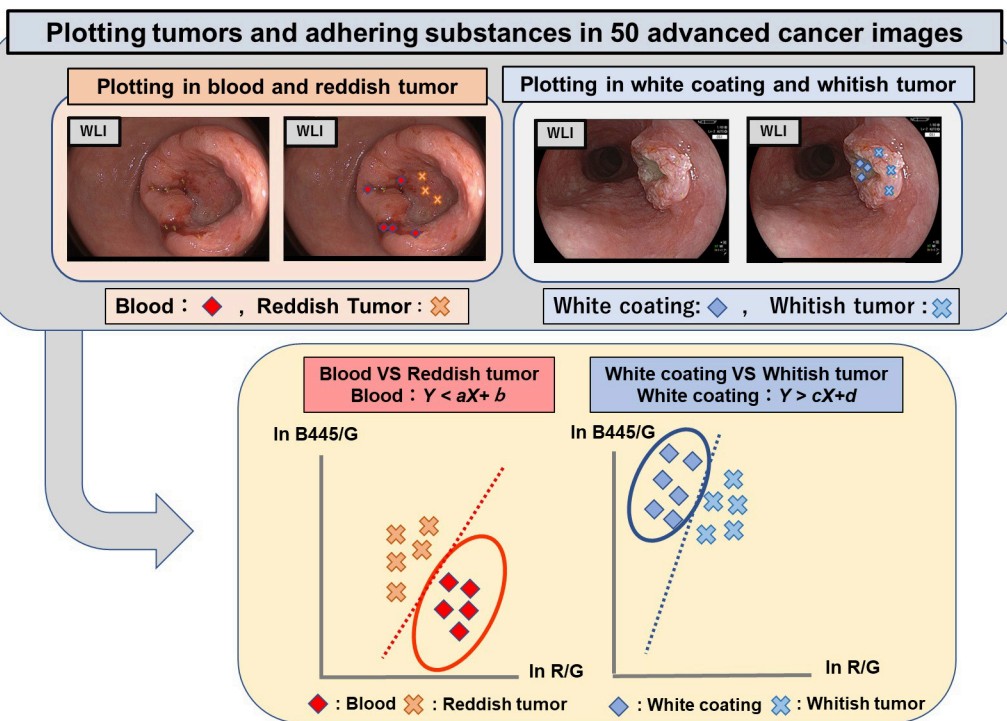

**Fig 2. Constructing scatter diagrams and discriminating algorithms.** WLI images of advanced cancers were used to generate plotted areas. Two endoscopists plotted areas considered to represent the exposed tumor and areas considered to represent adherent substances (blood and white coating) at multiple locations in each of the images of the 50 advanced cancers using ImageJ (open-source, public-domain image processing software). The plotted areas were identified by two-dimensional coordinates, the spectrum signal values (the raw pixel values proportional to the reflection rates of the substances concerned) at those areas were analyzed, and the two-dimensional scatter plots [Y axis: ln (B445/G), X axis: ln (R/G)] were produced.

concentration in the tissue under observation. The blue spectrum signal B445 is a narrow-band signal at the 445 nm wavelength that is strongly affected by hemoglobin absorbance. Because B445 is more strongly affected by the blood adhering to the tissue surface than the green spectrum signal (G), the value of ln (B445/G) would decrease. Due to these influences of blood on the signal intensity levels, we anticipated that hemorrhage should exhibit a positive correlation with the X axis and a negative correlation with the Y axis on two-dimensional scatter plots, whereas white coating devoid of blood should exhibit a negative correlation with the X axis and a positive correlation with the Y axis. The use of logarithmic spectrum signal values helps to increase the likelihood of a linear relationship between hemoglobin concentration and waveform in physical terms, with the aim of making it easier to use linear relationships defined by linear functions for discriminant analysis. Linear discriminant analysis was performed with CAnalysis Ver. 7.2 (https://www.heisei-u.ac.jp/ba/fukui/analysis.html) to derive discrimination formulae to differentiate between a tumor and its adherent substances (blood vs. reddish tumor; white coating vs. whitish tumor) on the scatter plots [Y axis: ln (B445/G), X axis: ln (R/G)].

## Validation analysis of discriminating algorithms

Finally, we evaluated discriminating algorithms as follows. Areas considered to constitute exposed tumor tissue and areas considered to constitute adherent substances were plotted on a total of 15 images (6 advanced gastric carcinomas and 9 advanced esophageal carcinomas) that were randomly selected from the 50 images of advanced cancers, and scatter plots [Y: ln

(B445/G); X axis: ln (R/G)] were produced as a validation data set. The diagnostic reliability of the discrimination formulae was then investigated using these validation scatter plots, and the StO2 values of the areas identified by the discriminating algorithms as tumor tissue or the adherent substances were analyzed. For examining the accuracy rates (AR) of the discriminating algorithms, cross-validation was performed twice by replacing the plot data for the validation set with the same number of plot data used for constructing algorithms selected using a random number table.

## Results

A total of 50 images of 50 lesions (20 advanced gastric cancers and 30 advanced esophageal cancers) were analyzed for constructing discriminating algorism. There is no significant difference in clinical and pathological characteristics between cases used in discriminant algorithms construction and validation (Table 1). The following areas were plotted in 50 tumors: 68 sites of blood; 83 sites of reddish tumor; 89 sites of white coating; and 79 sites of whitish tumor. Red, green, and blue spectrum signal intensity values were obtained from these selected sites, and it was shown that there are differences in the signal intensity values of blood and reddish tumor, as well as white coating and whitish tumor, which can be seen closely in endoscopic images (Table 2). To analyze the differences between the spectral signal intensity values of tumors and their adherent substances, scatter plots [Y axis: ln (B445/G); X axis: ln (R/G)] were produced, and analysis using linear discrimination formulae was used to calculate the algorithms that best discriminated between tumors and their adherent substances. And then, Linear discriminant analysis was performed with CAnalysis Ver. 7.2 to derive discrimination algorithms. The discriminating algorithm for blood vs. areas of reddish tumor was $Y > -4.90X + 7.13$ and that for white coating vs. areas of whitish tumor was $Y < -0.52X + 0.17$ (Fig 3). Plotting was performed for 15 images randomly selected from the 50 images of advanced cancers,

**Table 1. Clinicopathological findings.**

| | | Cases for constructing algorithm (50) | Cases for validation (15) |
|---|---|---|---|
| Sex | Male, n (%) | 35 (70) | 9 (60) |
| | Female, n (%) | 15 (30) | 6 (40) |
| Age, mean (SD) | | 62.9 (8.8) | 69.4 (8.2) |
| Cancer type | Gastric Cancer | 20 (40) | 6 (40) |
| | Esophageal Cancer | 30 (60) | 9 (60) |
| Size (mm), median [range] | | 55 [20–130] | 40 [20–130] |
| Lesion Type | Type1, n (%) | 5 (10) | 3 (20) |
| | Type2, n (%) | 36 (72) | 7 (47) |
| | Type3, n (%) | 9 (18) | 5 (33) |
| | Type4, n (%) | 0 (0) | 0 (0) |
| Clinical depth | cT2, n (%) | 6 (12) | 3 (20) |
| | cT3<, n (%) | 44 (88) | 12 (80) |
| Lymph node metastasis | Absent, n (%) | 4 (8) | 2 (13) |
| | Present n (%) | 46 (92) | 13 (87) |
| Distant metastasis | Absent n (%) | 14 (28) | 7 (47) |
| | Present n (%) | 36, (72) | 8 (53) |
| 8th edition of TNM | I, n (%) | 0 (0) | 0 (0) |
| | II, n (%) | 4 (8) | 2 (13) |
| | III, n (%) | 10 (20) | 5 (33) |
| | IV, n (%) | 36 (72) | 8 (53) |

**Table 2. Spectrum signal values in plotting areas of images for constructing algorithms.**

| Spectrum signal values in plotting areas of images for constructing algorithms | | | | |
| --- | --- | --- | --- | --- |
| | Plotting Points (n) | Red spectral signal value (average, median, [range]) | Green spectral signal value (average, median, [range]) | 445nm Blue spectral signal value (average, median, [range]) |
| Blood | 68 | 232.9, 188.8, [20.1–724.4] | 36.6, 30.4, [4.7–114.3] | 25.0, 18.7, [4.0–126.5] |
| Reddish tumor | 83 | 376.2, 346.1, [128.7–1057.2] | 118.0, 98.3, [41.6–457.2] | 91.5, 75.8, [33.0–523.6] |
| White coating | 89 | 398.1, 371.9, [126.3–899.3] | 266.8, 249.6, [63.4–713.8] | 226.0, 196.1, [53.4–733.1] |
| Whitish tumor | 79 | 380.2, 350.5, [127.6–779.5] | 211.4, 197.8, [67.9–559.6] | 212.8, 199.7, [64.8–631.8 ] |

containing 24 sites of blood, 25 sites of reddish tumor, 25 sites of white coating, and 25 sites of whitish tumor. The spectrum signal intensity values at these plotted sites were analyzed and scatter plots [Y axis: ln(B445/G); X axis: ln(R/G)] for validation were produced (Table 3). The spectrum signal intensity value at one of the sites with blood was very different from those at the other 24 sites and was excluded from the investigation as an outlier. The scatter plots in the validation set were then used to verify the diagnostic reliability of discrimination between the tumor and its adherent substances; the following results were obtained: 95.9% (discriminating algorithm for blood vs. reddish tumor) and 96.0% (discriminating algorithm for white coating vs. whitish tumor) (Table 4 and Fig 4). It was shown that blood and reddish tumors in endoscopic images, as well as white coating and whitish tumors can be distinguished with high accuracy. To evaluate how much the current OS imaging is influenced by adherent substances, StO2 values were examined in the three regions of the whole tumor region including adherent tumors, only tumor region, and only adherent substances region defined by the discriminating algorithms. The StO2 value (median, [range]) in the areas consisting of both blood and reddish tumor was 75.9% (37.8%–100.0%); the StO2 values of the area discriminated by the algorithm were 79.3% (37.8%–100.0%) for blood and 74.5% (62.0%–86.9%) for the exposed reddish tumor. The StO2 value measured in the areas consisting of both white coating and whitish tumor was 66.1% (42.1%–100.0%); the StO2 values measured in the areas discriminated by the

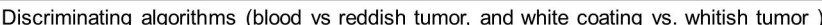

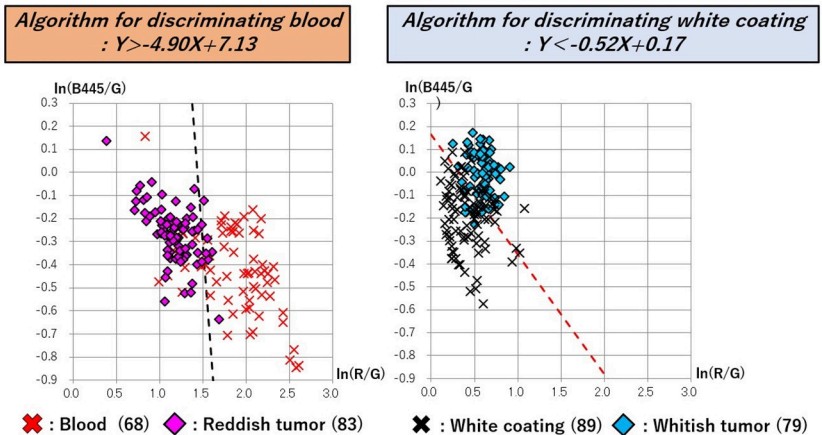

**Fig 3. Discriminating algorithms for the tumor vs. its adherent substances.** Discriminating algorithm for blood vs. areas of reddish tumor was Y> -4.90X + 7.13 and that for white coating vs. areas of whitish tumor was Y< -0.52X + 0.17. The scatter plots in the validation set were then used to verify the diagnostic reliability of the discrimination between the tumor and its adherent substances.

**Table 3. Spectrum signal values in plotting areas of images for validation set.**

| | Plotting Points (n) | Red spectral signal value (average, median, [range]) | Green spectral signal value (average, median, [range]) | 445nm Blue spectral signal value (average, median, [range]) |
|---|---|---|---|---|
| Blood | 24 | 302.5, 290.1, [132.2–619.4] | 36.7, 31.9, [18.0–57.6] | 24.2, 21.5, [12.7–43.1] |
| Reddish tumor | 25 | 462.0, 515.6, [169.4–655.5] | 134.3, 140.8, [51.0–208.3] | 96.5, 94.6, [41.2–168.1] |
| White coating | 25 | 476.6, 461.0, [154.6–955.9] | 310.1, 303.6, [100.2–547.6] | 242.0, 217.9, [73.3–532.1] |
| Whitish tumor | 25 | 441.3, 439.2, [224.9–682.2] | 232.1, 246.9, [121.0–340.9] | 219.2, 225.4, [94.4–382.6] |

algorithm were 73.8% (42.1%–100.0%) for white coating and 65.7% (53.0%–76.3%) for whitish tumor. The StO2 range distributions were narrower in the areas of tumors than in the areas of their adhered substances, and it was suggested that the current OS imaging may be strongly influenced by adherent substances (Fig 5). To verify the reproducibility of the accuracy rate of the discriminating algorithm, cross validation was performed twice. In the first analysis, the accuracy rates of the discriminating algorithms for blood vs. reddish tumor and white coating vs. whitish tumor were 83.7% and 82.0%, respectively (Fig 6). In the second analysis, the corresponding accuracy rates were 89.8% and 82.0%, respectively (Fig 7).

## Discussion

Diagnostic imaging methods targeting the evaluation of hypoxic environments in tumors have been attracting attention. Useful parameters, such as a biomarker for predicting the effects of anti-cancer agents of quantification of StO2 values in tumors, are important to determine therapeutic strategy. The gold standard method for the quantification of StO2 values is the direct measurement of StO2 in tumors using an Eppendorf needle electrode system [16]. However, due to the insertion of an electrode into a tumor, this method is highly invasive and difficult to perform accurately when a lesion is small or when it is located at a deeper site. Nuclear medicine approaches are attractive methods to quantify StO2. Positron emission tomography (PET)-based hypoxia imaging using various probes has been reported for assessing StO2 non-invasively. PET-based hypoxia imaging with probes such as FMISO [17, 18], 64Cu-ATSM [19], and 18F-FAZA [20] has been reported as a method for non-invasive estimation of StO2 levels. In the basic mechanism of action, a probe labeled with a radioisotope (RI) is accumulated within cells under the hypoxic condition, which allows visualization of the hypoxic tumor region and its quantitative analysis. By using this principle, hypoxic areas can be identified through PET Imaging [21]. Although PET-CT with a probe allows non-invasive estimation of StO2 values in tumors, it has a disadvantage of being affected by drug metabolism. In addition, it takes time for a probe to accumulate in a lesion; furthermore, the probe's

**Table 4. Accuracy rate of discriminating algorithms for the tumor vs. its adherent substances.**

| Algorithm for discriminating blood: $Y > -4.90X + 7.13$ | | | |
|---|---|---|---|
| | Positive | Negative | Total |
| Blood | 24 | 0 | 24 |
| Reddish tumor | 2 | 23 | 25 |
| | | | Accuracy rate (%): 95.9 |

| Algorithm for discriminating white coating: $Y > -0.52X + 0.17$ | | | |
|---|---|---|---|
| | Positive | Negative | Total |
| White coating | 25 | 0 | 25 |
| Whitish tumor | 2 | 23 | 25 |
| | | | Accuracy rate (%): 96.0 |

Verifying the diagnostic reliability of discriminating algorithms

**Algorithm for discriminating blood** : $Y > -4.90X + 7.13$

**Algorithm for discriminating white coating** : $Y < -0.52X + 0.17$

✖ : Blood (24)  ◆ : Reddish tumor (25)  ◆ : Whitish tumor (25)  ✖ : White coating (25)

**Fig 4. Examining accuracy rates of discriminating algorithms for the tumor vs. its adherent substances.** The scatter plots in the validation set were then used to verify the diagnostic reliability of the discrimination between the tumor and its adherent substances: 95.9% (discriminating algorithms for blood vs. reddish tumor) and 96.0% (discriminating algorithms for white coating vs. whitish tumor).

metabolism and elimination pathway are visualized as well, which impedes the diagnosis of the lesion. PET imaging also requires expensive instrumentation that are set in a controlled radiation area.

There are possibly three characteristic points of OS imaging compared with other modalities capable of measuring StO2 in tumors used in our practice for cancer patients. Firstly, similar to PET-based hypoxia imaging, the intestinal mucosal StO2 levels can be non-invasively estimated without puncturing, which is considered an important factor for evaluating StO2 levels in routine clinical settings. Furthermore, drug administration is not required for OS imaging, which is a major strength of this method compared to other modalities. Secondly, assessment with other image-enhanced endoscopic modalities such as Blue Laser Imaging (BLI) [22, 23] and Linked Color Imaging (LCI) [24, 25] can be easily performed concomitantly

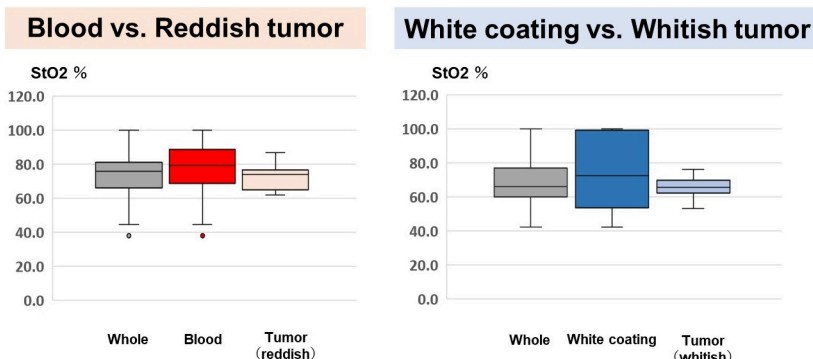

**Fig 5. Analysis of StO2 values of the areas identified by the discriminating algorithms.** StO2 values (median, [range]) of the areas consisting of both blood and reddish tumor was 75.9% (37.8%–100.0%); the StO2 values of the area discriminated by the algorithm were 79.3% (37.8%–100.0%) for blood and 74.5% (62.0%–86.9%) for the exposed reddish tumor. The StO2 value measured in the areas consisting of both white coating and whitish tumor was 66.1% (42.1%–100.0%); the StO2 values measured in the areas discriminated by the algorithm were 73.8% (42.1%–100.0%) for the white coating and 65.7% (53.0%–76.3%) for the whitish tumor.

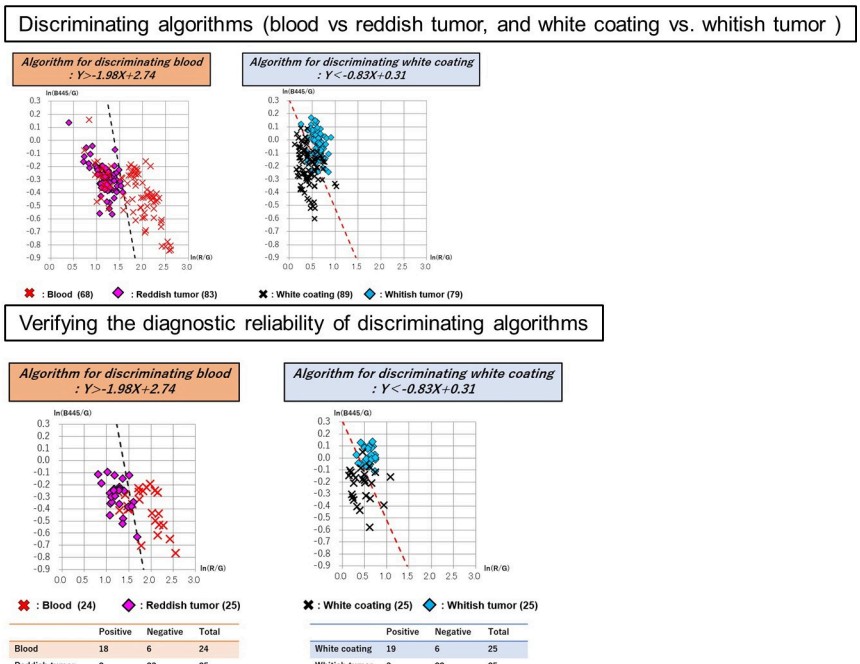

**Fig 6. In first cross validation, discriminating algorithms for tumors vs. their adhered substances were identified and examination of the algorithm's diagnostic reliability was performed.** In the first cross validation, the discriminating algorithm for blood vs. areas of reddish tumor was Y> -1.98X + 2.74 and that for white coating vs. areas of whitish tumor was Y< -0.83X + 0.31. The scatter plots in the validation set were then used to verify the diagnostic reliability of discrimination between the tumor and its adherent substances: 83.7% (discriminating algorithm for blood vs. reddish tumor) and 82.0% (discriminating algorithm for white coating vs. whitish tumor).

with OS imaging, which allows for total qualitative assessment. Finally, StO2 levels can be measured in real- time. The real-time estimation of StO2 levels can be particularly useful as it allows targeted biopsy based on tumor biological character. This aspect reveals an additional advantage of endoscopic therapy as the effect of the additional treatment can be determined instantly. This approach is especially useful for judging the effect of endoscopic treatment, such as photodynamic therapy [26, 27], RFA [28], and cryoablation for esophageal cancer [29].

While OS imaging is a very useful clinical application, it has certain limitations. The measurement of StO2 levels is difficult in some cases of advanced GI cancer due to adherent substances such as blood or white coating, covering the surface of tumors. The expectation is that the OS imaging can be used as a diagnostic method in the future to accurately estimate StO2 levels of the entire tumor. This is an essential approach when assessing the efficacy of chemotherapy or endoscopic therapy such as PDT. However, it would be essential to remove the noise derived from the adherent substances. Our algorithms allow the use of OS imaging to differentiate between the StO2 levels measured in surface tissue layers and those measured in the tumor area covered with adherent substances. The current OS imaging can evaluate the StO2 value for each pixel in the endoscopic imaging, but it means that the StO2 information of the adherent substances is included when evaluating the StO2 values of the whole tumor.

The construction of the discriminating algorithms will make it possible to evaluate tumor-specific StO2 values by automatically removing the StO2 information in the region defined by the algorithm as adherent substances.

Our algorithms showed a high rate of discrimination between the tumor and its adherent substances: 95.9% (blood vs. reddish tumor) and 96.0% (white coating vs. whitish tumor).

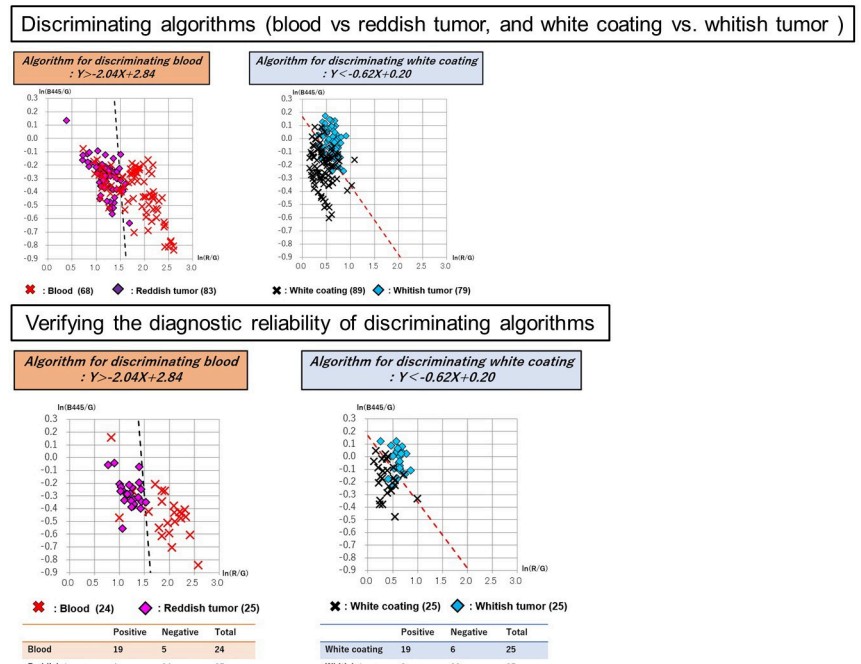

**Fig 7. In a second cross validation, discriminating algorithms for tumors vs. their adhered substances were identified and examination of the algorithm's diagnostic reliability was performed.** In the second analysis of cross validation, the discriminating algorithm for blood vs. areas of reddish tumor was Y> -2.04X + 2.84 and that for white coat vs. areas of whitish tumor was Y< -0.62X + 0.20. The scatter plots in the validation set were then used to verify the diagnostic reliability of discrimination between the tumor and its adhering substances: 89.8% (discriminating algorithm for blood vs. reddish tumor) and 82.0% (discriminating algorithm for white coating vs. whitish tumor).

During cross validation, the accuracy rate decreased, but the results were reproducible. It can be concluded that the algorithm can accurately discriminate between the tumor and its adhered substances.

In recent years, methods of image processing that use deep learning approaches have been reported in several studies. It would be important to consider a more detailed image processing analysis for obtaining accurate StO2 information on tumors [30]. One of the limitations of this study is that a relatively small number of tumor images were used and annotation points of this study during image analysis were selected optionally by only two endoscopists. While satisfactory results have been obtained using the verification data, annotating the noise and lesion area with more images and additional analysis of the whole area is being considered. Another limitation is that StO2 level was measured to construct algorism in small samples of heterogeneous cluster of advanced cancers with various clinical characteristics. While there is no significant difference in clinical pathological characteristics between tumors used in discriminant algorithms construction and tumors used for verification, the possibility that differences in clinical pathological characteristics of tumors affect StO2 levels is not considered in this study.

Finally, this study is focused on blood and white coating on the surface of tumors as a common representative source of noise; however, in routine clinical settings, halation, saliva, or digestive juices can also contribute to the noise. We plan to work on further applications of OS imaging, including the analysis of other types of clinical information to establish the automated optimal StO2 measuring tool for this novel endoscopy system.

If algorithms for removing a lot of noises can be constructed and implemented in the current OS imaging system, it will be possible to clarify the clinical significance of measuring the

StO2 values of the tumor surface. In addition, it will be possible to evaluate changes in the tumor during treatment by constructing the algorism that removes noise automatically. Oxygen saturation imaging can be expected to play a role as a real time biomarker for predicting or determining the effect of anti-cancer treatments including PDT or RFA when conquering the influence of adherent substances with introduction of our proposed algorism.

## Conclusions

OS imaging is strongly influenced by the presence of adherent substances; therefore, it is important to eliminate the noise caused by adherent substances for evaluating the specific StO2 value of tumors. We developed a novel approach to measure StO2 levels, specifically in tumors. Our algorithms allow the differentiation between StO2 levels that originate specifically from the tumor tissue and those that are derived from the tumor adherent substances, such as blood and white coating. Our approach brings endoscopy imaging to the new level that will help improve cancer diagnostics and treatment.

## Supporting information

**S1 File.**
(XLS)

## Author Contributions

**Conceptualization:** Keiichiro Nishihara, Keisuke Hori, Takaaki Saito, Toshihiko Omori, Hironori Sunakawa, Tatsunori Minamide, Masayuki Suyama, Yoichi Yamamoto, Yusuke Yoda, Kensuke Shinmura, Hiroaki Ikematsu, Tomonori Yano.

**Data curation:** Keiichiro Nishihara, Keisuke Hori, Takaaki Saito, Masayuki Suyama, Yoichi Yamamoto, Yusuke Yoda, Kensuke Shinmura, Hiroaki Ikematsu, Tomonori Yano.

**Formal analysis:** Keiichiro Nishihara, Takaaki Saito.

**Funding acquisition:** Tomonori Yano.

**Investigation:** Keiichiro Nishihara, Keisuke Hori, Takaaki Saito.

**Methodology:** Keiichiro Nishihara, Keisuke Hori, Takaaki Saito, Toshihiko Omori, Hironori Sunakawa, Tatsunori Minamide, Masayuki Suyama, Yoichi Yamamoto, Yusuke Yoda, Hiroaki Ikematsu, Tomonori Yano.

**Project administration:** Keiichiro Nishihara, Keisuke Hori, Takaaki Saito, Toshihiko Omori, Tatsunori Minamide, Tomonori Yano.

**Resources:** Keiichiro Nishihara, Keisuke Hori, Hironori Sunakawa, Yoichi Yamamoto, Yusuke Yoda, Tomonori Yano.

**Software:** Keiichiro Nishihara, Keisuke Hori, Takaaki Saito.

**Supervision:** Keisuke Hori, Toshihiko Omori, Hironori Sunakawa, Tatsunori Minamide, Masayuki Suyama, Yusuke Yoda, Kensuke Shinmura, Hiroaki Ikematsu, Tomonori Yano.

**Validation:** Keiichiro Nishihara, Keisuke Hori, Takaaki Saito, Yusuke Yoda, Tomonori Yano.

**Visualization:** Keiichiro Nishihara, Keisuke Hori, Takaaki Saito, Yusuke Yoda, Tomonori Yano.

**Writing – original draft:** Keiichiro Nishihara.

**Writing – review & editing:** Keisuke Hori, Takaaki Saito, Toshihiko Omori, Hironori Suna-
kawa, Tatsunori Minamide, Masayuki Suyama, Yoichi Yamamoto, Yusuke Yoda, Kensuke
Shinmura, Hiroaki Ikematsu, Tomonori Yano.

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
