## [Decision Letter · Decision Letter 0]

6 Aug 2020

PONE-D-20-12468

A study of evaluating specific tissue oxygen saturation values of gastrointestinal tumors by removing adherent substances in oxygen saturation imaging

PLOS ONE

Dear Dr. Yano,

Thank you for submitting your manuscript to PLOS ONE. After careful consideration, we feel that it has merit but does not fully meet PLOS ONE’s publication criteria as it currently stands. Therefore, we invite you to submit a revised version of the manuscript that addresses the points raised during the review process.

We look forward to receiving your revised manuscript.

Kind regards,

Qinghui Zhang

Academic Editor

PLOS ONE

Journal Requirements:

2. In your ethics statement in the Methods section and in the online submission form, please provide additional information about the images used in your study. Please disclose the source of these images, whether they can be publicly accessed, and whether images were fully anonymized and de-identified before you accessed them. If not, please state whether the IRB or ethics committee waived the requirement for informed consent or whether patients gave written or verbal consent. If consent was verbal, please describe how consent was documented and witnessed.

"Tomonori Yano received a research grant from FUJIFILM; however, this was not a conflict of interest. Oxygen saturation imaging was provided by FUJIFILM Corporation for the clinical study. Other authors have no conflicts of interest."

We note that one or more of the authors have an affiliation to the commercial funders of this research study: FUJIFILM Corporation.

3.1. Please provide an amended Funding Statement declaring this commercial affiliation, as well as a statement regarding the Role of Funders in your study. If the funding organization did not play a role in the study design, data collection and analysis, decision to publish, or preparation of the manuscript and only provided financial support in the form of authors' salaries and/or research materials, please review your statements relating to the author contributions, and ensure you have specifically and accurately indicated the role(s) that these authors had in your study. You can update author roles in the Author Contributions section of the online submission form.

3.2. Please also provide an updated Competing Interests Statement declaring this commercial affiliation along with any other relevant declarations relating to employment, consultancy, patents, products in development, or marketed products, etc. 

Reviewers' comments:

Reviewer's Responses to Questions

**Comments to the Author**

1. Is the manuscript technically sound, and do the data support the conclusions?

Reviewer #1: Partly

Reviewer #2: Yes

2. Has the statistical analysis been performed appropriately and rigorously? 

Reviewer #1: N/A

Reviewer #2: Yes

3. Have the authors made all data underlying the findings in their manuscript fully available?

Reviewer #1: No

Reviewer #2: Yes

4. Is the manuscript presented in an intelligible fashion and written in standard English?

Reviewer #1: No

Reviewer #2: Yes

5. Review Comments to the Author

Reviewer #1: In this paper Nishihara and colleagues discuss about the efficacy of a new algorithm to evaluate specific tissue oxygen saturation values of gastrointestinal tumors by removing adherent substances in oxygen saturation imaging. The paper topic is potentially interesting because it highlights the possibility of a new, less invasive method to discriminate precancerous or cancerous areas in GI lesions using the superficial tissue oxygen saturation (StO2) and plotting different tumor areas to differentiate neoplastic tissue from other confounding variables such as adherent substances (blood clot ect).

Hovewer even if the topic could be interesting there are some major concerns about the paper content that could be considered before eventual pubblication:

1) In material and methods there is no specification about tumor type, site, size, grading and general stadiation (T and N stage by CT scan or eus or PET). There is a table but these fundamental informations are not integrated in the text and this can be misleading. Moreover the authors do not specify if this heterogeneous cluster of lesions can be matched or could affect the results they discuss later in the text.

2) the entire result section is almost confusing. It's not specify how the authors utilize the algorithm, how they decide to apply it (indications by the Fujifilm i.e?) and in general the results are showed using the figures but they are not clearly explained so it's difficult to read this part.

3) the abstract section has to be re written starting from the over mentioned point.

4) the discussion is also not to clear because the authors do not specify how the results can be used in the clinical practice probably it's due to the unclear results section too.

Reviewer #2: As mentioned by the authors, the main limits of this study are the limited number of patients and that the annotation point were selected randomly and by only two endoscopist. This could affect the reliability of this algorithm, that should be validated in a larger multi-center study in order to evaluate the reproducibility of this algorithm.

However, this study try to overcome the limitations of this new technology, which could have interesting applications in gastrointestinal tumors in the next future.

I suggest to the authors to largely explain in the discussion the possible future applications on this technology and largely explain the differences with other diagnostic imaging methods.

6. PLOS authors have the option to publish the peer review history of their article (what does this mean?). If published, this will include your full peer review and any attached files.

Reviewer #1: No

Reviewer #2: No

---

## [Author Response · Author response to Decision Letter 0]

17 Oct 2020

Dear reviewers

We thank you for giving us the opportunity to revise the manuscript. We have carefully studied all the suggestions. The manuscript has benefited from these insightful suggestions. We look forward to working with you and the reviewers to move this manuscript closer to publication in PLOS ONE. The manuscript has been rechecked and the necessary changes have been made in accordance with the reviewer’s suggestions. 

The responses to all comments have been prepared and given below.

Thank you for your consideration. We look forward to hearing from you

---

## [Decision Letter · Decision Letter 1]

17 Nov 2020

A study of evaluating specific tissue oxygen saturation values of gastrointestinal tumors by removing adherent substances in oxygen saturation imaging

PONE-D-20-12468R1

Dear Dr. Yano,

We’re pleased to inform you that your manuscript has been judged scientifically suitable for publication and will be formally accepted for publication once it meets all outstanding technical requirements.

Kind regards,

Qinghui Zhang

Academic Editor

PLOS ONE

Additional Editor Comments (optional):

Reviewers' comments:

Reviewer's Responses to Questions

**Comments to the Author**

1. If the authors have adequately addressed your comments raised in a previous round of review and you feel that this manuscript is now acceptable for publication, you may indicate that here to bypass the “Comments to the Author” section, enter your conflict of interest statement in the “Confidential to Editor” section, and submit your "Accept" recommendation.

Reviewer #1: All comments have been addressed

2. Is the manuscript technically sound, and do the data support the conclusions?

Reviewer #1: Yes

3. Has the statistical analysis been performed appropriately and rigorously? 

Reviewer #1: Yes

4. Have the authors made all data underlying the findings in their manuscript fully available?

Reviewer #1: Yes

5. Is the manuscript presented in an intelligible fashion and written in standard English?

Reviewer #1: Yes

6. Review Comments to the Author

Reviewer #1: I think that the authors modified the paper in the proper manner, addressing clearly all the points eraised in the previous review. Congrats

7. PLOS authors have the option to publish the peer review history of their article (what does this mean?). If published, this will include your full peer review and any attached files.

Reviewer #1: No

---

## [Editor Report · Acceptance letter]

28 Dec 2020

PONE-D-20-12468R1 

A study of evaluating specific tissue oxygen saturation values of gastrointestinal tumors by removing adherent substances in oxygen saturation imaging 

Dear Dr. Yano:

I'm pleased to inform you that your manuscript has been deemed suitable for publication in PLOS ONE. Congratulations! Your manuscript is now with our production department. 

Kind regards, 

on behalf of

Dr. Qinghui Zhang 

Academic Editor

PLOS ONE